

# Intranodal signal suppression in pelvic MR lymphography of prostate cancer patients: a quantitative comparison of ferumoxtran-10 and ferumoxytol

Oscar A. Debats[1], Ansje S. Fortuin[1,2], Hanneke J.M. Meijer[3], Thomas Hambrock[1], Geert J.S. Litjens[1], Jelle O. Barentsz[1] and Henkjan J. Huisman[1]

[1] Department of Radiology and Nuclear Medicine, Radboudumc, Nijmegen, The Netherlands
[2] Department of Radiology, Ziekenhuis Gelderse Vallei, Ede, The Netherlands
[3] Department of Radiation Oncology, Radboudumc, Nijmegen, The Netherlands

## ABSTRACT

**Objectives**. The key to MR lymphography is suppression of T2* MR signal in normal lymph nodes, while retaining high signal in metastatic nodes. Our objective is to quantitatively compare the ability of ferumoxtran-10 and ferumoxytol to suppress the MR signal in normal pelvic lymph nodes.

**Methods**. In 2010, a set of consecutive patients who underwent intravenous MR Lymphography (MRL) were included. Signal suppression in normal lymph nodes in T2*-weighted images due to uptake of USPIO (Ultra-Small Superparamagnetic Particles of Iron Oxide) was quantified. Signal suppression by two USPIO contrast agents, ferumoxtran-10 and ferumoxytol was compared using Wilcoxon's signed rank test.

**Results**. Forty-four patients were included, of which all 44 had a ferumoxtran-10 MRL and 4 had additionally a ferumoxytol MRL. A total of 684 lymph nodes were identified in the images, of which 174 had been diagnosed as metastatic. USPIO-induced signal suppression in normal lymph nodes was significantly stronger in ferumoxtran-10 MRL than in ferumoxytol MRL ($p < 0.005$).

**Conclusions**. T2* signal suppression in normal pelvic lymph nodes is significantly stronger with ferumoxtran-10 than with ferumoxytol, which may affect diagnostic accuracy.

## INTRODUCTION

Prostate cancer (PCa) is the most common type of cancer and the second leading cause of cancer death in men (*Siegel, Miller & Jemal, 2015*). The presence of lymph node metastases is a poor prognostic factor, reducing treatment options. Conventional imaging modalities such as CT or MRI rely on size and shape criteria to detect metastatic lymph nodes, resulting in poor sensitivity and specificity (*Hövels et al., 2008*; *Heesakkers et al., 2008*). Pelvic lymph node dissection (PLND) is currently regarded as the gold standard for lymph node staging in PCa patients, but comes with increased costs and risk of morbidity (*Heidenreich et al., 2011*;

Corresponding author
Oscar A. Debats, debats@gmail.com

Loeb, Partin & Schaeffer, 2010), and not all lymph node metastases are found at routine PLND (Heesakkers et al., 2009). To select patients for PLND or elective nodal irradiation, various nomograms (Partin et al., 1993; Partin et al., 1997) and numerical formulae (Roach et al., 1994; Nguyen et al., 2009) are used to predict nodal involvement. However, these do not provide information on the number, size, and location of metastatic nodes, which are important parameters for staging (Cheng et al., 2012). 11C-choline PET/CT has been shown to be more accurate than CT and MR (Schiavina et al., 2008), but has limited sensitivity in the substantial group of smaller lymph node metastases <7 mm (Fortuin et al., 2012).

MR Lymphography (MRL) outperforms 11C-choline PET/CT by providing good accuracy for lymph nodes well below 7 mm. MRL uses a specific, Ultra-Small Superparamagnetic Particles of Iron Oxide (USPIO) based contrast agent. USPIOs accumulate in macrophages in normal lymphatic tissue, resulting in signal suppression on T2*-weighted MRI. Normal lymph nodes become dark, and when fat-saturation is applied, fade into the background of the surrounding dark fat. Thus, metastatic lymph nodes stand out with bright signal intensity (SI) (Harisinghani et al., 1999). MRL properly performed and interpreted can provide a high negative predictive value (NPV) (95–99%). A substantial amount of prostate cancer patients can thus be spared a PLND procedure and the associated risk of morbidity.

Two MRL USPIO contrast agents can be currently used for humans. MRL using ferumoxtran-10 (Combidex®; SPL Medical, Nijmegen, The Netherlands) is the only prospectively investigated imaging modality for assessing metastatic involvement of pelvic lymph nodes, with sensitivities up to 91%, at 98% specificity (Harisinghani et al., 2003). Despite these encouraging results, ferumoxtran-10 has not yet reached the market. Currently, ferumoxtran-10 is produced in The Netherlands by SPL Medical B.V. under GMP (good manufacturing practice) conditions. The pharmacy of our institution takes care of the supply of ferumoxtran-10 (for clinical routine and for clinical trials) for its patients in accordance with the Dutch law.

The iron replacement drug ferumoxytol (Landry et al., 2005) (Feraheme®, AMAG Pharmaceuticals Inc, Cambridge, MA) has been proposed as a potential alternative MRL contrast agent. In 2007, Harisinghani et al. (2007) concluded that ferumoxytol MRL potentially identifies malignant lymph nodes. To the best of our knowledge however, no prospective studies have yet been performed to validate the off-label use of this drug as an MRL contrast agent. Moreover, the FDA has issued a warning considering its off-label use, and explicitly recommends intravenous administration by a slow drip infusion: In the safety announcement, issued on 30 March 2015, which is available on the website www.fda.gov, it is stated that all IV iron products carry a risk of potentially life-threatening allergic reactions, and that a Boxed Warning had been added to the prescribing instructions of ferumoxytol that describes these serious risks and specifies a number of recommendations. One of these recommendations is 'Only administer diluted Feraheme as an IV infusion over a minimum of 15 min. Feraheme should not be given as an undiluted IV injection.'

The purpose of this pilot study was to quantitatively compare ferumoxytol and ferumoxtran-10 for use in MRL. The outcome measure was signal suppression in normal lymph nodes, as this is the basis for discriminating metastatic and normal ones.

**Table 1  Scan parameters for Magnetic Resonance Lymphography.**

| Name | Description | Imaging plane | Echo time (ms) | Repetition time (ms) | Flip angle (deg) | Pixel size (mm) | Matrix | Slice thickness (mm) | Slices | Bandwidth (Hz/pixel) |
|---|---|---|---|---|---|---|---|---|---|---|
| VIBE | T1-weighted spin echo | Coronal | 2.45 | 4.95 | 10 | 0.8 × 0.8 | 320 × 320 | 0.8 | 240 | 400 |
| MEDIC | T2*-weighted gradient echo | Coronal | 11 | 20 | 10 | 0.8 × 0.8 | 320 × 320 | 0.8 | 240 | 180 |

## MATERIALS AND METHODS

This study retrospectively evaluated clinically obtained MRL data. The scientific use of clinically obtained image data was approved by the Institutional Review Board. All patients provided written informed consent for the use of the obtained data for research purposes.

### Patient selection

Between January and April 2010, a set of consecutive patients who had been referred to the Radboud university medical center, Nijmegen for clinically indicated MRL were included in this retrospective study. The inclusion criteria were: (1) a histologically confirmed prostate cancer with intermediate to high risk for nodal metastases; (2) a ferumoxytol MRL and/or ferumoxtran-10 MRL performed between January and April 2010; (3) successfully acquired 3D T1-weighted (''VIBE'') and 3D T2*-weighted (''MEDIC'') sequences available.

### MRL protocol

The MRL protocol was as follows. USPIO contrast was administered intravenously, 36 to 24 h before the MRI was performed. Because of this time interval, only post-contrast images were acquired. The same time interval was applied for both types of MRL. For ferumoxtran-10 MRL, this has been established as the optimal interval. For ferumoxytol MRL, this was considered optimal by our expert readers based on visual inspection of a set of nine MRLs acquired from three patients at different time intervals post injection. One patient was imaged at day 0 (directly post injection), at day 1, and at day 2, and the other two were imaged at day 0, day 1, and day 3.

Immediately before imaging, Buscopan (20 mg i.v. and 20 mg i.m.) and Glucagon (20 mg i.m.) were administered to suppress bowel peristalsis. The dose of ferumoxtran-10 was 2.6 mg Fe per kg body weight, conform earlier research (*Harisinghani et al., 2003*). The dose of ferumoxytol was 6.0 mg Fe per kg body weight. This is the maximum allowed dose, which was chosen to maximize the signal suppression in the MEDIC images.

Imaging was performed using a 3.0 T MR-scanner (Magnetom TrioTim; Siemens, Erlangen, Germany). Scan parameters are listed in Table 1.

### Interpretation of MRL images

The radiological diagnosis of the MRL examinations was established as the consensus reading by two expert readers: an MD specially trained in reading MRL scans (OD, 2 years of MRL experience, >300 MRLs), and an abdominal radiologist (JB, >10 years of MRL experience, >1,000 MRLs). The T1-weighted sequences (see Table 1) were used

for localization and assessment of shape and size of the lymph nodes, and the iron-sensitive T2*-weighted sequences were used to assess USPIO uptake, as described by *Heesakkers et al. (2008)*.

## Quantitative MRL analysis

The contrast uptake in all detected lymph nodes was computed as follows. All lymph nodes visible in the pelvic region were interactively segmented using the computer application *Lymph Node Task Card* (Siemens, Malvern, PA). Segmentation was performed based on the T1-weighted images, in which all lymph nodes (normal as well as metastatic ones) appear as hypointense structures. From these three-dimensional segmentations, the volume of each lymph node was recorded automatically.

As a measure of contrast uptake, relative SI was computed rather than absolute SI. Lymph node assessment based on relative SI (local fat calibrated lymph node assessment) is more similar to visual assessment of the MRL image by a radiologist: visual assessment is also based on a comparison of the SI of a lymph node with the SI of the surrounding fat. Local fat calibration compensates for coil profiles and other factors creating a spatially varying SI distortion in the images. The fat calibration was implemented by manually segmenting a region of fatty tissue in the direct neighborhood of each lymph node. The relative SI is calculated by subtraction of the mean SI of the local fatty tissue region from the SI in the corresponding lymph node.

## Statistical analysis

Relative SI in normal lymph nodes was compared using Wilcoxon's signed rank test for paired non-parametric data, using the statistical software package SPSS (version 20.0). $P$-values >0.05 were considered statistically significant. Box-and-whisker plots were constructed using the R Environment for Statistical Computing.

## RESULTS

Forty-four patients fulfilled the inclusion criteria. All patients had prostate cancer staged as Gleason Score 6 or higher. All underwent ferumoxtran-10 MRL, and four patients also underwent ferumoxytol MRL. In all cases, ferumoxtran-10 MRL was performed first. The time intervals between the two types of MRL for these four patients were as follows: 23, 130, 240, and 241 days (0.8, 4.3, 7.9 and 7.9 months) respectively. Median prostate specific antigen (PSA) and Gleason Score were 9.5 (range 0.01–954.0) and 7 (range 6–9), respectively. A total of 684 lymph nodes were identified in the MRL images, and 57 (8.3%) of those belonged to patients who received both types of MRL. All 684 nodes were found to be suitable for quantification of signal suppression. The readers diagnosed 174 lymph nodes (25%) as positive (i.e., metastatic).

Median size (volume) of the positive lymph nodes was 0.14 ml (inter-quartile range (IQR) 0.043–0.45) and median volume of the negative lymph nodes was 0.12 ml (IQR 0.045–0.30). A histogram of lymph node sizes is shown in Fig. 1.

Visual evaluation of the MEDIC images demonstrated that in ferumoxtran-10 MRL the signal of normal lymph nodes was markedly suppressed; thus they were generally as dark

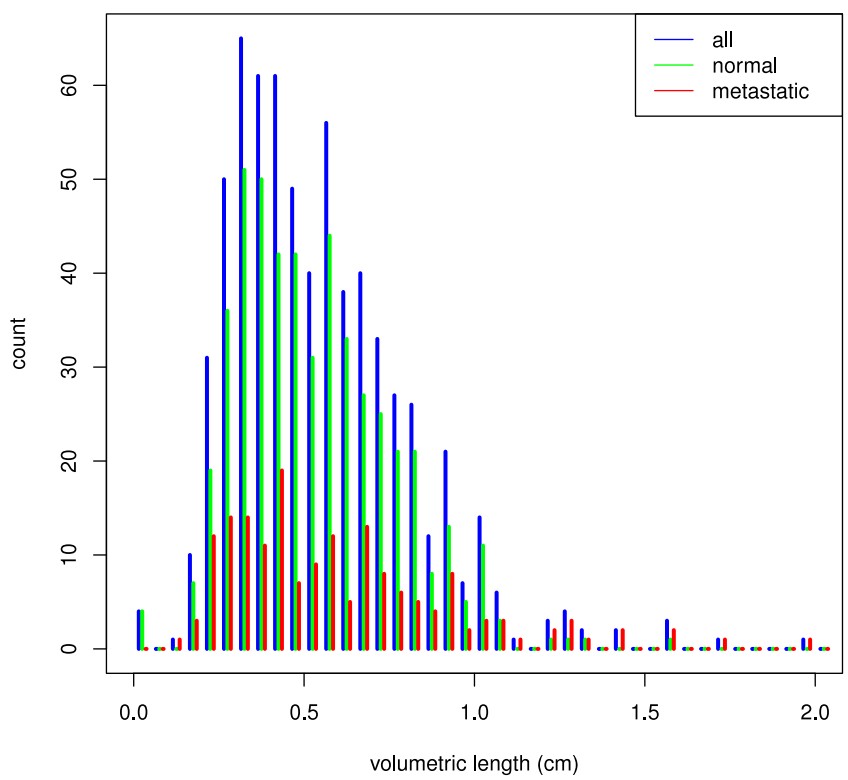

**Figure 1** **Histogram of lymph node size.** The volumetric length of a lymph node is defined as the cubic root of its volume.

as the surrounding (suppressed) fatty tissue, whereas metastatic nodes remained visible as bright structures. With ferumoxytol, normal nodes also had suppressed signal but the suppression was less than with ferumoxtran-10, and they remained brighter than the fatty tissue background, still apparent as hyperintense structures (Fig. 2).

The signal intensity box plot (Fig. 3) shows that with ferumoxtran-10, the interquartile range (IQR) of normal lymph node intensity had an overlap with the IQR of fatty tissue intensity. However, with ferumoxytol, the IQR of normal lymph nodes did not overlap with the IQR of fatty tissue.

The difference in relative SI of normal lymph nodes between the two types of MRL was significant. Relative SI was on average 39.7 (95% confidence interval (CI) [31.1, 48.3]) in ferumoxytol MRL, and $-2.1$ (95% CI [$-8.0$–3.8]) in ferumoxtran-10 MRL ($p < 0.005$).

## DISCUSSION

The results of this pilot study show, that relative SI in normal lymph nodes was significantly higher in post-contrast ferumoxytol MRL than in ferumoxtran-10 MRL, both visually and quantitatively ($p < 0.005$). The MRL protocol used in this study included only post-contrast imaging. However, as pre-contrast SI (whether absolute or relative) is not influenced, of course, by the choice of contrast agent, pre-contrast SI would not differ between ferumoxytol MRL and ferumoxtran-10 MRL, and any significant difference in relative SI

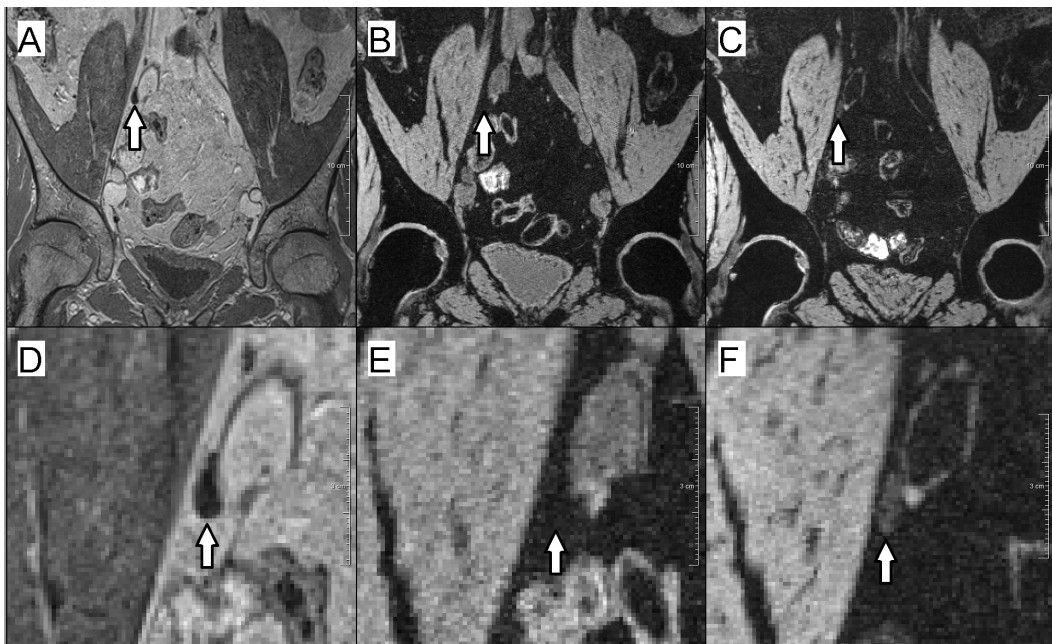

**Figure 2** **Example of a normal lymph node in ferumoxtran-10 MRL and ferumoxytol MRL.** (A–C) Overviews; (D–F) zoomed-in images. (A & D) USPIO-insensitive 3D T1-weighted (VIBE) sequence. The lymph node is visible as a hypointense structure. (B & E) 3D T2*-weighted (MEDIC) sequence, enhanced with ferumoxtran-10. The normal lymph node is as dark as the fat-suppressed fat and is thus indistinguishable from the background. (C & F) 3D T2*-weighted sequence, enhanced with ferumoxytol. Due to contrast uptake, the normal lymph node is darker than it would have been in non contrast-enhanced MRI, but it is not as dark as the background, and thus may be scored as metastatic.

refers to a difference in signal suppression. In other words, the difference in relative SI found in our analysis implies that signal suppression was weaker for ferumoxytol MRL than for ferumoxtran-10 MRL.

The underlying mechanism causing the difference in signal suppression in normal lymph nodes is not known. Possibly it can be explained by the different coating of the particles, which may lead to different uptake by macrophages, or different clearance. A limited number of studies have been published comparing ferumoxtran-10 and ferumoxytol as MR contrast agent. Interestingly, both agents appear suitable for detection of macrophages in atherosclerotic plaques (*Herborn et al., 2006*), and for detection of antigen-induced arthritis (*Simon et al., 2006*). A possible explanation for the difference in behavior when used for lymph node imaging may be the selective uptake of ferumoxtran-10, but not ferumoxytol, by macrophages that migrate specifically to nodal tissue. This might be caused by the different coatings of ferumoxtran-10, or by the different hydrodynamic diameter. In a recent study comparing three USPIO's in a porcine model, it was shown that differences in hydrodynamic diameter were associated with significant differences in lymphatic iron accumulation (*Pouw et al., 2015*).

The MR sequences used in this study have been validated in earlier research (*Heesakkers et al., 2008*). One might argue that the lesser signal suppression in ferumoxytol MRL could be improved by optimizing the sequences to make them more sensitive to USPIO.

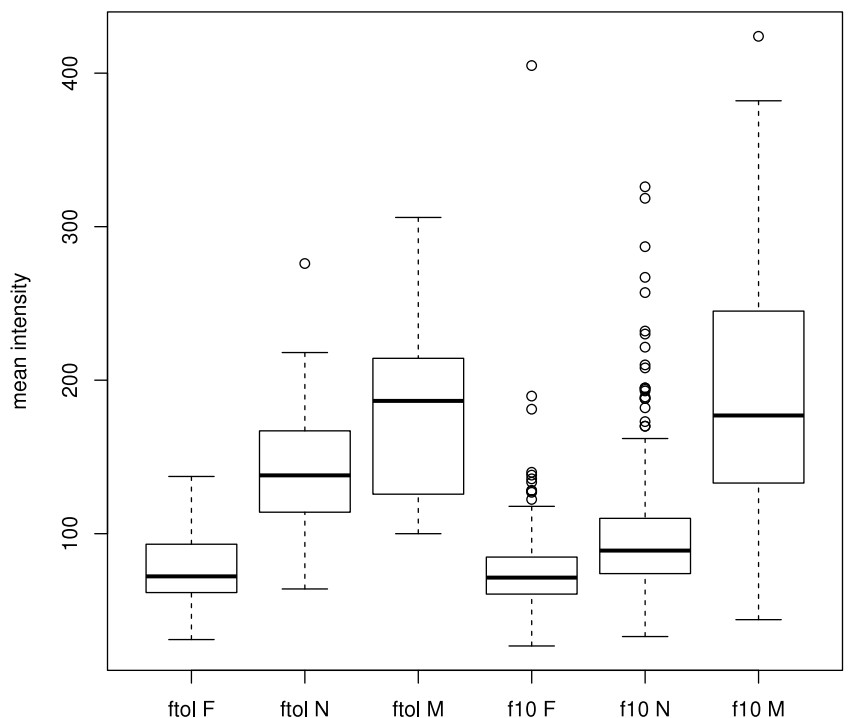

**Figure 3** **Tukey box plot comparing the signal intensities of lymph nodes and fatty tissue regions for both contrast agents.** F, fatty tissue regions; N, normal lymph nodes; M, metastatic lymph nodes; ftol, ferumoxytol; f10, ferumoxtran-10.

However, this would lead to an increase of artifacts related to bowel peristalsis, which would deteriorate image quality.

The vast majority of the lymph nodes in our data set are of normal size (i.e., with short-axis >10 diameter). As can be seen in Fig. 1, the distribution of lymph node size does not differ substantially between positive and negative lymph nodes. This is in accordance with the results reported by *Tiguert et al. (1999)*, who analyzed 980 prostatectomy patients and concluded that in normal-sized lymph nodes, size did not correlate with the presence of metastasis.

This study has some limitations. In this pilot study, the number of patients was limited: only four patients received both types of MRL. However, by performing a quantitative analysis on a nodal level in the same set of node-by-node compared lymph nodes we were able to perform a valid analysis of the differences. A total of 684 nodes, of which 57 were imaged with both contrast agents, were analyzed, and demonstrated a significant difference in intranodal signal suppression between the two contrast agents. In future studies, these results need to be confirmed by including a larger number of patients with a ferumoxytol MRL.

Another limitation is that the optimal time interval—24 to 36 h—between contrast administration and imaging has only been investigated qualitatively for ferumoxytol MRL. As described in the Methods section, this interval was considered optimal based on visual inspection of a set of ferumoxytol MRLs acquired after different time intervals, but a quantitative investigation is needed to create a firmer basis for this choice.

Radiologists who want to use ferumoxytol as an alternative to ferumoxtran-10 should be aware that MRL with ferumoxytol as a contrast agent needs to be interpreted differently even with identical MR sequences. In ferumoxtran-10 MRL, a lymph node that remains bright is highly suspicious for metastasis, but this clear separation does not apply for ferumoxytol MRL. Thus, the distinction between normal and metastatic nodes is less straightforward. This is illustrated in Fig. 3, where the box plots of *ferumoxytol-normal* and *ferumoxytol-metastatic* overlap substantially, which does not occur with ferumoxtran-10. Thus, using ferumoxytol may lead to either many more false positives, or much lower sensitivity. Considering the already relatively low PPV of 69% of ferumoxtran-10 (*Heesakkers et al., 2008*), a significant further decrease in PPV with ferumoxytol may be problematic and lead to diagnostic inaccuracy or uncertainty. This is important because ferumoxtran-10 is available on a limited scale, whereas ferumoxytol can be used off-label. Thus, knowledge about the advantages and disadvantages of ferumoxytol is crucial.

In conclusion: USPIO-induced signal suppression in normal lymph nodes is significantly weaker for ferumoxytol than for ferumoxtran-10, and therefore its discriminative performance is likely to be lower. Therefore, the successful results reported in previous studies regarding ferumoxtran-10 MRL cannot be extrapolated to ferumoxytol MRL. Ferumoxytol MRL, when used in an off-label mode as a replacement for ferumoxtran-10 MRL, is likely to result in more false positives, and should not be used in clinical practice for the diagnosis of metastatic lymph nodes without further scientific validation.

### Funding

This project was funded by Grant KUN2007-3971 from the Dutch Cancer Society. The funders had no role in study design, data collection and analysis, decision to publish, or preparation of the manuscript.

### Grant Disclosures

The following grant information was disclosed by the authors:
Dutch Cancer Society: KUN2007-3971.

### Competing Interests

SPL Medical Nijmegen produces and owns Combidex; however, none of the authors have any financial relationship with SPL Medical or a conflict of interest. Henkjan J. Huisman is an Academic Editor for PeerJ.

### Author Contributions

- Oscar A. Debats and Henkjan J. Huisman conceived and designed the experiments, performed the experiments, analyzed the data, contributed reagents/materials/analysis tools, wrote the paper, prepared figures and/or tables, reviewed drafts of the paper.
- Ansje S. Fortuin analyzed the data, wrote the paper, reviewed drafts of the paper.

- Hanneke J.M. Meijer, Jelle O. Barentsz and Thomas Hambrock wrote the paper, reviewed drafts of the paper.
- Geert J.S. Litjens contributed reagents/materials/analysis tools, wrote the paper, reviewed drafts of the paper.

## Human Ethics

The following information was supplied relating to ethical approvals (i.e., approving body and any reference numbers):

Institutional review board: CMO Regio Arnhem-Nijmegen. This study retrospectively evaluated clinically obtained MRL data. The scientific use of clinically obtained image data was approved by the Institutional Review Board.

## Data Availability

The raw data has been supplied as Data S1.

## Supplemental Information

Supplemental information for this article can be found online at http://dx.doi.org/10.7717/peerj.2471#supplemental-information.

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
