# Peer review of "Intranodal signal suppression in pelvic MR lymphography of prostate cancer patients: a quantitative comparison of ferumoxtran-10 and ferumoxytol"

_PeerJ, doi:10.7717/peerj.2471_

## Round 0.1 · original submission · Minor Revisions

Dear Authors,

I have read the manuscript and am in agreement with the 2 reviewers that it should only need Minor Revisions.

As noted by reviewer 2, the 2 materials which are used under 2 very different legal routes, this needs a more detailed explanation of the legal use in humans.

·

Basic reporting

•the submission adhere to all PeerJ policies
•professional standards of English language
•well structured article
•relevant figures
•the submission include all results relevant to the hypothesis

Experimental design

•primary research in a retrospective analysis
•clearly defined research question
• high technical standard of investigation procedure
•methods are well described
.

Validity of the findings

• robust and statistically sound data
• uncontrolled data (no comparison with histological findings...)
•the conclusion is connected to the original question investigated
•speculation about the different enhancement patterns of the 2 presented CMs is identified as such

Additional comments

# " dose of ferumoxytol was 6.0 mg Fe per kg body ": why is the concentration of Fe higher in ferumoxytol; please explain...
# did you compare your MRL results with PLND findings?
# " the availability of ferumoxtran However is limited because it is currently not approved. ": please explain

Reviewer 2 ·

Basic reporting

OK

Experimental design

The quality of the results is a little hampered by the small number of subject for the ferumoxytol case (only 4).

Line 97-98 needs a clear ref.

Which T1 method is used in line 107.

Line 166, this is investigated in pigs and described by: J. Pouw Int. J. of nanomedicine 2015. 10, P 1235.

Line 171-172: The second coating (PEG) is not mentioned.

Validity of the findings

A clear presentation of all the obtained data is missing in particular for the ferumoxytol case. In particular with no significant statistics, simply presenting all observed data is the best thing to do. In particular on line 93-95 these data are missing. In this case it is not clear which patients are considered here.

In line 132-133 the time-point of the measurement is not clear.
- What is the order & timing of the administered agent and the MRI
- Can you comment on the washout time of both agents

Additional comments

The current trend (line 50) is an opinion not supported by facts or lit.

The two materials are used under two very different legal routes, this needs a more detailed explanation of the legal use in humans.

The safety issue on line 72-74 is not clear for a general reader, please explain with more detail

---

## Round 0.2 · accepted · Accept

The revision has been well performed.

Reviewer 2 ·

Basic reporting

The changes made by the authors are fine to my opinion.

Experimental design

OK

Validity of the findings

OK